

# Impact of maternal iron and zinc intake on low birth weight risk: a nested case—control study

Liping Yang[1,2,*], Zifu Wang[2,*], Lei Cao[2], Yuqing Li[2], Jingyan Wang[2], Shuyu Ding[2] and Baohong Mao[3]

[1] Department of Public Health and Infection Management, Gansu Provincial Hospital of Traditional Chinese Medicine, Gansu, Lanzhou, China
[2] School of Public Health, Gansu University of Chinese Medicine, Gansu, Lanzhou, China
[3] Department of Medical Education, Gansu Provincial Maternity and Child-care Hospital, Gansu, Lanzhou, China
[*] These authors contributed equally to this work.

## ABSTRACT

**Background.** The association between maternal iron/zinc intake and the risk of low birth weight (LBW) in infants is still unclear. This study aimed to investigate the effects of iron and zinc daily intake preconception and during pregnancy on the risk of delivering LBW babies and to assess whether there is an interaction between iron and zinc intake on the risk of LBW.

**Methods.** A nested case–control study was performed, including 565 cases and 7,510 controls in Lanzhou City, China. Eligible participants were interviewed about their diets and characteristics during pregnancy. Unconditional logistic regression was used to determine the association between dietary iron/zinc intake and the risk of LBW and its clinical subtypes. Multivariate-adjusted restricted cubic spline (RCS) models were applied to explore the nonlinear association between dietary iron/zinc intake levels and the risk of LBW.

**Results.** Lower intake of iron/zinc preconception and during pregnancy was associated with a higher risk of LBW and some subtypes, and there was a nonlinear trend between iron/zinc intake and the occurrence risk of LBW ($P_{\text{Nonlinear}} < 0.05$). Moreover, the synergistic effects of low iron and low zinc intake on the risk of LBW were found.

**Conclusion.** Efforts to promote iron and zinc intake preconception and during pregnancy need to be strengthened to reduce the local incidence of LBW.

## INTRODUCTION

According to the World Health Organization (WHO) criterion (*De Costa et al., 2021*), low-birth-weight (LBW) infants are defined as newborns weighing less than 2,500 g in the first hour after birth, independent of gestational age. Because of immature development and many complications, LBW is closely related to neonatal death, developmental delays, and long-term chronic diseases in offspring (*Shinzawa et al., 2019*). Studies have shown that LBW accounts for 15–20% of all live births and more than 80% of all neonatal deaths

Corresponding author
Baohong Mao, Baohong.Mao@gszy.edu.cn

worldwide; it is still an important public health indicator for assessing the health of infants and young children (*Blencowe et al., 2019*). The overall prevalence of LBW of Chinese children younger than 6 years was 5.15% in 2013, with 4.57% in boys and 5.68% in girls (*Shen et al., 2021*). However, the potential mechanisms for LBW remain unclear. Therefore, identifying modifiable risk factors to provide evidence for the primary prevention of LBW is particularly meaningful.

Maternal nutritional status during pregnancy is an important modifiable factor associated with their health and affects the next generation's growth and development. Adequate nutrition particularly important for the growth, development, and maintenance of normal physiological functions in LBW infants is mainly obtained from the diet. Micronutrient deficiency occurs if the body has insufficient reserves preconception and insufficient supplies after pregnancy. The main function of trace element iron is to transport oxygen and carbon dioxide, participate in tissue respiration, and promote biological redox reactions (*Berggren et al., 2015*; *Tussing-Humphreys et al., 2012*). Zinc is the second most abundant trace element in the body after iron; it is especially abundant in the liver, kidney, retina, bone, and muscle tissue and fluid, and is involved in the regulation of cell function and vitamin A metabolism in the body (*Dusek, Jankovic & Le, 2012*). Zinc deficiency in fetuses can lead to growth restriction and sexual organ dysfunction (*Khoushabi et al., 2016*).

In recent years, the scientific community has been widely concerned about the biological effects of various trace elements during pregnancy and their relationship with the origin of life, heredity, growth and development, congenital disabilities, and other maternal and infant health outcomes. However, available studies on the relationship between iron or zinc intake during pregnancy and LBW have not systematically covered the period preconception and the entire pregnancy (*Chang, Li & Xu, 2018*; *Siyoum & Melese, 2019*; *Palma et al., 2008*; *Yadav et al., 2020*; *Imdad & Bhutta, 2012*; *Haider et al., 2013*; *Cogswell et al., 2003*; *Peña Rosas et al., 2015*; *Bánhidy et al., 2011*; *Negandhi et al., 2014*; *Iannotti et al., 2008*; *Xie, Chen & Pan, 2001*; *Goldenberg et al., 1995*; *Zahiri Sorouri, Sadeghi & Pourmarzi, 2016*; *Osendarp et al., 2000*; *Carducci, Keats & Bhutta, 2021*; *Warthon-Medina et al., 2015*). Some of the studies that have assessed total iron/zinc intake did not separately model the relationships between iron/zinc from food and that from dietary supplements. Moreover, previous studies have not considered the synergistic effects of iron and zinc intake on the risk of LBW. Therefore, this study aimed to investigate the effects of iron and zinc daily intake preconception and during pregnancy on the risk of delivering LBW babies and to assess whether there is an interaction between iron and zinc intake on the risk of LBW.

# MATERIALS & METHODS

## Study population

A birth cohort study was conducted from January 2018 to June 2019 at Gansu Provincial Maternity and Child Care Hospital, the largest hospital of its kind in Lanzhou City, China. Eligible study participants were pregnant women with no history of mental illness, 18 years old or older, and those who had regular prenatal care. Written informed consent was obtained from all participants. A total of 8,879 eligible women were approached for

participation. After removing the missing information on neonatal weight and dietary survey during pregnancy, stillbirth, and multiple births, a total of 8,673 pregnant women with single live births were finally included. Among them, 7,510 women had normal-birth-weight (NBW) infants, 565 had LBW infants, and 598 had macrosomic infants. The Institutional Review Board of Gansu Provincial Maternity and Child Care Hospital approved this study [No. 2018(029)].

### Dietary surveys, iron, and zinc daily intake

Daily surveys and dietary iron and zinc intake were collected through a face-to-face semiquantitative food frequency questionnaire (FFQ) (*Liu et al., 2016*). The survey included sociological characteristics, disease history, physiological fertility, and dietary survey. The dietary status was investigated by a 24-h dietary recall survey administered by trained medical staff. The types and quantities of all kinds of food consumed in three consecutive days were investigated, which mainly included 12 categories of cereals, oils and fats, vegetables, fruits, poultry, livestock meat and its products, eggs, aquatic products, beans and soy products, milk and milk products, fungi and algae, snacks and drinks, and other 59 kinds of common food.

This study is a nested case-control study based on the previous birth cohort. The dietary nutrition survey for each research subject was conducted four times. The first time was during the preconception consultation and health check-up within 3 months preconception, after obtaining informed consent, a preconception dietary nutrition survey was conducted. The second time was during the first trimester ($\leq$ 13 weeks of gestation) when going for prenatal check-ups at the hospital. The third and fourth times were during the second trimester (14–27 weeks of gestation) and the third trimester ($\geq$ 28 weeks of gestation) when going for prenatal check-ups at the hospital again, respectively. After completing all dietary surveys, the daily intake of dietary vitamins and trace elements of each pregnant woman during different pregnancies was calculated following the second edition of the Chinese Food Composition Table 2009 (*Yang, Wang & Pan, 2009*). Daily vitamin and trace element intakes in different pregnancies were divided into groups based on the recommended nutrient intake (RNI) of Chinese residents (*National Health Commission of the People's Republic of China, 2017*) for statistical analysis. The RNI is the amount of a nutrient that is enough to ensure that the needs of 97.5% of the population are met. Data on pregnancy-related complications and birth outcomes were extracted from medical records.

### LBW

LBW infants (*World Health Organization, 2004*), defined as newborns weighing less than 2,500 g in the first hour after birth, were included in the case group, while infants with body weight greater than or equal to 2,500 g and lower than 4,000 g were defined as NBW infants and were included in the control group for analyses. LBW infants were further classified as preterm LBW infants (defined as newborns weighing less than 2,500 g in the first hour after birth and gestational age below 37 weeks) and term LBW infants (defined as newborns weighing less than 2,500 g in the first hour after birth and gestational age of more than or equal to 37 weeks).

## Body mass index and gestational weight gain

Preconception weight was self-reported during the first prenatal care visit. Body mass index (BMI) was calculated as weight (kg) divided by the square of height (m), and then subcategorized in accordance with the Working Group of Obesity in China as follows: underweight (BMI <18.5 kg/m$^2$), normal weight (18.5 kg/m$^2$ $\leq$ BMI <24 kg/m$^2$), overweight (24 kg/m$^2$ $\leq$ BMI <28 kg/m$^2$), and obesity (BMI $\geq$ 28 kg/m$^2$). Gestational weight gain in kg was calculated by subtracting preconception weight from maternal weight at delivery, which was categorized based on the US Institute of Medicine (IOM) GWG Guidelines 2009 (*Rasmussen, Yaktine & Institute of Medicine (US) and National Research Council (US) Committee to Reexamine IOM Pregnancy Weight Guidelines, 2009*).

## Covariates

The study covariates were as follows: maternal age, preconception BMI (kg/m$^2$), weight gain during pregnancy (kg), father's BMI (kg/m$^2$), total energy intake (kcal/d), dietary iron intake (mg/d), dietary zinc intake (mg/d), maternal ethnicity (Han/Minority), monthly income (<3,000 RMB per capita/$\geq$3,000 RMB per capita), maternal education level (college or above/below college), smoking include passive and active (passive smoking exposure was operationally defined as inhaling secondhand smoke from others' tobacco products for a minimum of 15 min daily) (Yes/No), maternal employ (Yes/No), multivitamin supplement (Yes/No), gestational hypertension (Yes/No), history of premature birth (Yes/No), reproductive history (Yes/No).

## Statistical analysis

Portions of this text were previously published as part of a preprint (*Huang et al., 2024*). Comparisons of the selected characteristics between women with LBW and NBW were evaluated using the chi-square test or Fisher's exact test. The differences in measurement data between the two groups were analyzed by independent-sample t test, and the variables that did not meet the normal distribution were compared between groups by the Wilcoxon rank-sum test. Unconditional logistic regression was used to determine the odds ratios (ORs) and 95% confidence intervals (CIs) for the association between dietary iron and zinc intake and the risk of LBW and its clinical subtypes (preterm LBW and term LBW). Confounding factors, including weight gain during pregnancy, father's BMI, total energy intake, monthly income per capita, maternal education level, smoking, maternal employment, multivitamin supplementation, gestational hypertension, history of premature birth, and reproductive history, were adjusted for in the unconditional logistic regression models. Dietary iron and zinc intake were categorized into quartiles, and the dose–response relationship (*P* for trend) was calculated based on those categorical levels. The association between dietary iron/zinc intake levels and the risk of LBW may result in a nonlinear correlation. Multivariate-adjusted restricted cubic spline (RCS) models with three knots were applied to explore the nonlinear association. The number of knots was determined by comparing the criteria between Bayesian and Akaike information. The RCS models were adjusted for the potential confounding factors listed above. The multiplicative interaction parameters (OR = OR11/(OR01× OR10)) and 95% CI were also estimated

by including the abovementioned variables. The interactions on the additive scale were assessed using relative excess risk due to interaction (RERI = RR11 − RR10 − RR01 + 1), attributable proportion (AP = RERI/RR11), and synergy index [S = (RR11 − 1)/[(RR01 − 1) + (RR10 − 1) (*Andersson et al., 2005*). We estimated 95% CI for each of these measures; the null values of relative excess risk due to interaction (RERI) and attributable proportion (AP) were 0, whereas the null value for S was 1. All statistical tests were two-sided. Analyses were performed using SAS 9.4 (SAS Institute, Inc., Cary, NC, USA). The RCS models were performed by R software, version 4.1.3 (package 'Hmisc', 'rms', and 'survival').

## RESULTS

### Basic characteristics of the study population

A total of 8,673 women were eligible for the final analysis. Among them, 565 had LBW infants, comprising 126 term LBW and 439 preterm LBW infants. As illustrated in Table 1, compared with women who had NBW infants, mothers of LBW infants were more likely to have lower monthly income, lower education level, less maternal employment, higher proportion of gestational hypertension, history of premature birth, and reproductive history. Moreover, case mothers had a significantly lower weight gain during pregnancy, lower total energy intake, fewer multivitamin supplements, and lower dietary iron and zinc intake. Distribution of the maternal ethnicity, drinking during pregnancy, gestational diabetes, anemia during pregnancy, and history of miscarriage did not significantly differ between the two groups.

### Associations of maternal dietary iron intake with the odds of LBW

As shown in Table 2, demonstrates a dose–response relationship between prepregnancy dietary iron intake and low birth weight (LBW) odds. Compared with quartile 4 (highest intake reference group), the adjusted odds ratios (95% confidence intervals) for LBW were: 1.33 (0.98−1.79) for quartile 3, 1.20 (1.03−1.41) for quartile 2 and 1.14 (1.02−1.27) for quartile 1 (lowest intake group), and the test for trend was significant ($P = 0.006$). Quartile 1 and quartile 2 of iron intake showed a significantly higher risk of preterm LBW compared with the highest quartile, and the test for trend was significant ($P = 0.007$). After stratifying by periods of pregnancy, an increased risk of LBW was observed for quartile 1 and quartile 2 compared with quartile 4, first trimester 1.16 (1.01−1.32) and 1.21 (1.02−1.41), Second trimester 1.18 (1.03−1.37) and 1.21 (1.03−1.44), Third trimester 1.17 (1.02−1.34) and 1.28 (1.08−1.52), and the test for trend was significant ($P_{\text{first trimester}} = 0.007$, $P_{\text{second trimester}} = 0.004$, and $P_{\text{third trimester}} = 0.007$). Similarly, the increased risk of preterm LBW was observed for quartile 1 and quartile 2 compared with quartile four preconception and stratifying by periods of the trimester of pregnancy. In addition, an increased risk of term LBW was only observed for quartile 1 compared with quartile four during the second trimester.

Mothers with iron intake below RNI had a significantly higher risk of LBW and preterm LBW preconception, during pregnancy, during the first trimester, and during the second trimester. Specifically, the adjusted OR for LBW was 1.26 (1.03–1.54), 1.13 (1.01–1.27), 1.25 (1.02–1.55), and 1.29 (1.04–1.60), while that for preterm LBW was 1.35 (1.08–1.69),

**Table 1   Distributions of selected characteristics of the study population.**

| Characteristics | NBW ($n = 7,510$) | LBW ($n = 565$) | $P$-value |
|---|---|---|---|
| Maternal age | $28.55 \pm 4.20$ | $28.53 \pm 5.26$ | 0.944 |
| Preconception BMI (kg/m$^2$) | $20.57 \pm 2.65$ | $20.80 \pm 3.01$ | 0.084 |
| Weight gain during pregnancy (kg) | $17.08 \pm 5.29$ | $13.55 \pm 5.93$ | <0.001 |
| Father's BMI (kg/m$^2$) | $23.84 \pm 3.08$ | $23.33 \pm 3.16$ | <0.001 |
| Total energy intake (kcal/d) | 1,686 (1,440, 1,970) | 1,537 (1,240, 1,843) | <0.001 |
| Dietary iron intake (mg/d) | 24.24 (19.59, 30.46) | 21.74 (17.29, 27.19) | <0.001 |
| Dietary zinc intake (mg/d) | 8.40 (6.71, 10.39) | 7.17 (5.19, 9.37) | <0.001 |
| Maternal ethnicity | | | 0.051 |
|     Han | 6,914 (92.06) | 507 (89.73) | |
|     Minority | 596 (7.94) | 58 (10.27) | |
| Monthly income (RMB per capita) | $2,996 \pm 1,372$ | $2,433 \pm 1,334$ | <0.001 |
| Maternal education level | | | <0.001 |
|     <College | 4,512 (60.08) | 447 (79.11) | |
|     ≥ College | 2,998 (39.92) | 118 (20.89) | |
| Smoking (passive and active) | | | 0.002 |
|     No | 6,013 (80.07) | 421 (74.51) | |
|     Yes | 1,497 (19.93) | 144 (25.49) | |
| Drink during pregnancy | | | 0.235[*] |
|     No | 7,499 (99.85) | 563 (99.64) | |
|     Yes | 11 (0.15) | 2 (0.36) | |
| Maternal employ | | | <0.001 |
|     No | 3,348 (44.58) | 332 (58.76) | |
|     Yes | 4,162 (55.42) | 233 (41.24) | |
| Multivitamin supplement | | | <0.001 |
|     No | 1,436 (19.12) | 191 (33.80) | |
|     Yes | 6,074 (80.88) | 374 (66.20) | |
| Gestational diabetes | | | 0.503[*] |
|     No | 7,451 (99.21) | 562 (99.47) | |
|     Yes | 59 (0.79) | 3 (0.53) | |
| Gestational hypertension | | | <0.001 |
|     No | 7,241 (96.42) | 444 (78.58) | |
|     Yes | 269 (3.58) | 121 (21.42) | |
| Anemia during pregnancy | | | 0.803 |
|     No | 6,698 (89.19) | 502 (88.85) | |
|     Yes | 812 (10.81) | 63 (11.15) | |
| History of miscarriage | | | 0.535 |
|     No | 6,477 (86.24) | 482 (85.31) | |
|     Yes | 1,033 (13.76) | 83 (14.69) | |
| History of premature birth | | | <0.001 |
|     No | 7,473 (99.51) | 537 (95.04) | |
|     Yes | 37 (0.49) | 28 (4.96) | |

**Table 1** (*continued*)

| Characteristics | NBW ($n = 7,510$) | LBW ($n = 565$) | *P*-value |
|---|---|---|---|
| Reproductive history | | | <0.001 |
| Primipara | 5,632 (74.99) | 343 (60.71) | |
| Multiparous | 1,878 (25.01) | 222 (39.29) | |
| Newborn's sex | | | 0.053 |
| Male | 3,893 (51.84) | 269 (47.61) | |
| Female | 3,617 (48.16) | 296 (52.39) | |

**Notes.**
*Fisher exact test.

1.14 (1.01–1.30), 1.27 (1.00–1.60), 1.28 (1.01–1.64), and 1.35 (1.02–1.80), respectively. However, no significant associations of iron intake below RNI with LBW and term LBW were observed for the third trimester (Table 3).

Figure 1 shows the RCS curves for the associations between iron intake and the risk of LBW preconception and during pregnancy. As the daily dietary iron intake increases, the risk of LBW decreases rapidly and then gradually increases. In the third trimester, when the intake is below 24.27 mg/d, the risk of LBW increases. When the intake is between 24.27 mg/d and 61.68 mg/d, iron intake is a protective factor for LBW occurrence. When the intake is greater than 61.68 mg/d, the risk of LBW increases. Similar results were obtained in preconception, first trimester and second trimester. respectively ($P_{\text{Nonlinear}} < 0.05$).

## Associations of maternal dietary zinc intake with the odds of LBW

As shown in Table 4, there was a positive relationship between zinc intake and the risk of LBW. Compared with quartile 4 (highest) and quartile 1 (lowest) of dietary zinc intake during pregnancy and at the third trimester, the adjusted OR for LBW was 1.22 (1.01–1.46) and 1.20 (1.01–1.43), respectively, and the test for trend was significant for both ($P < 0.001$). Additionally, in quartile 1 of zinc intake during pregnancy, the second and third trimesters showed a significantly higher preterm LBW risk than in quartile 4, and the trend test was significant. However, no significant association was observed between zinc intake and term LBW either preconception or when stratifying by the trimesters of pregnancy. There was no relationship between zinc intake below RNI and the risk of LBW, preterm LBW, and term LBW (Table 5).

Figure 2 depicts the RCS curves for the associations between zinc intake and the risk of LBW preconception and during pregnancy. As the daily dietary zinc intake increases, the risk of LBW decreases rapidly and gradually levels off. In the third trimester, when the intake is less than 8.37 mg/d, the risk of LBW increases. When the intake is between 8.37 mg/d and 41.63 mg/d, zinc intake is a protective factor for LBW. When the intake is greater than 41.63 mg/d, the risk of LBW increases. Similar results were obtained in preconception, the first trimester, and the second trimester. respectively ($P_{\text{Nonlinear}} < 0.05$).

## Interaction effects of maternal dietary iron and zinc intake on the odds of LBW

We further analyzed the interaction effects of maternal dietary iron and zinc intake on the risk of LBW. An additive interaction between low iron and zinc dietary intake and LBW

Yang et al.
2025
10.7717/peerj.19896

**Table 2** Associations of maternal dietary iron intake with the odds of LBW.

| Dietary iron intake (mg/d) | NBW (7,510) | Cases | OR[a] (95% CI) | P-value | OR[b] (95% CI) | P-value | Cases | OR[b] (95% CI) | P-value | Cases | OR[b] (95% CI) | P-value |
|---|---|---|---|---|---|---|---|---|---|---|---|---|
| | | | LBW (n = 565) | | | | Term-LBW (37≥weeks) | | | Preterm-LBW (<37 weeks) | | |
| **Preconception** | | | | | | | | | | | | |
| Q1 <15.98 | 1,878 | 182 | 1.21 (1.12–1.32) | <0.001** | 1.14 (1.02–1.27) | 0.019* | 37 | 1.01 (0.81–1.26) | 0.930 | 145 | 1.15 (1.02–1.30) | 0.024* |
| Q2 15.98–20.16 | 1,877 | 154 | 1.23 (1.08–1.40) | <0.001** | 1.20 (1.03–1.41) | 0.023* | 33 | 1.11 (0.82–1.50) | 0.498 | 121 | 1.23 (1.04–1.47) | 0.019* |
| Q3 20.16–25.77 | 1,878 | 127 | 1.24 (0.95–1.63) | 0.110 | 1.33 (0.98–1.79) | 0.064 | 32 | 1.42 (0.79–2.52) | 0.236 | 95 | 1.27 (0.90–1.79) | 0.177 |
| Q4 ≥25.77 | 1,877 | 102 | Ref. | | Ref. | | 24 | Ref. | | 78 | Ref. | |
| P for trend | | | <0.001 | | 0.006 | | | 0.785 | | | 0.007 | |
| **During pregnancy** | | | | | | | | | | | | |
| Q1 <19.59 | 1,878 | 213 | 1.28 (1.18–1.40) | <0.001** | 1.20 (1.05–1.37) | 0.007** | 46 | 1.26 (0.97–1.64) | 0.084 | 167 | 1.18 (1.01–1.37) | 0.033* |
| Q2 19.59–24.24 | 1,877 | 140 | 1.18 (1.05–1.33) | 0.009** | 1.18 (1.00–1.40) | 0.054 | 34 | 1.18 (0.86–1.62) | 0.305 | 106 | 1.20 (0.99–1.45) | 0.061 |
| Q3 24.24–30.46 | 1,877 | 114 | 1.17 (0.90–1.52) | 0.284 | 1.21 (0.88–1.66) | 0.239 | 22 | 0.96 (0.51–1.81) | 0.899 | 92 | 1.28 (0.90–1.83) | 0.173 |
| Q4 ≥ 30.46 | 1,878 | 98 | Ref. | | Ref. | | 24 | Ref. | | 74 | Ref. | |
| P for trend | | | <0.001 | | 0.003 | | | 0.166 | | | 0.005 | |
| **First trimester** | | | | | | | | | | | | |
| Q1 <18.94 | 1,878 | 205 | 1.27 (1.17–1.38) | <0.001** | 1.16 (1.01–1.32) | 0.030* | 46 | 1.24 (0.92–1.67) | 0.157 | 159 | 1.15 (1.01–1.33) | 0.046* |
| Q2 18.94–23.64 | 1,877 | 150 | 1.22 (1.07–1.39) | 0.003** | 1.21 (1.02–1.41) | 0.021* | 38 | 1.16 (0.91–1.48) | 0.232 | 112 | 1.21 (1.00–1.45) | 0.044* |
| Q3 23.64–29.52 | 1,877 | 109 | 1.08 (0.82–1.43) | 0.589 | 1.07 (0.78–1.46) | 0.423 | 17 | 0.70 (0.36-1.37) | 0.295 | 92 | 1.19 (0.84–1.69) | 0.330 |
| Q4 ≥29.52 | 1,878 | 101 | Ref. | | Ref. | | 25 | Ref. | | 76 | Ref. | |
| P for trend | | | <0.001 | | 0.007 | | | 0.086 | | | 0.017 | |
| **Second trimester** | | | | | | | | | | | | |
| Q1 <19.67 | 1,878 | 209 | 1.30 (1.19–1.41) | <0.001** | 1.18 (1.03–1.37) | 0.024* | 45 | 1.35 (1.02–1.77)* | 0.033* | 164 | 1.19 (1.02–1.38) | 0.024* |
| Q2 19.67–24.46 | 1,877 | 143 | 1.23 (1.07–1.40) | 0.003** | 1.21 (1.03–1.44) | 0.026* | 34 | 1.27 (0.91–1.76) | 0.156 | 109 | 1.21 (1.01–1.47) | 0.046* |
| Q3 24.46–30.79 | 1,877 | 118 | 1.24 (0.94–1.64) | 0.125 | 1.23 (0.90–1.71) | 0.206 | 26 | 1.35 (0.72-2.55) | 0.352 | 92 | 1.20 (0.85–1.72) | 0.310 |
| Q4 ≥30.79 | 1,878 | 95 | Ref. | | Ref. | | 21 | Ref. | | 74 | Ref. | |
| P for trend | | | <0.001 | | 0.004 | | | 0.141 | | | 0.006 | |
| **Third trimester** | | | | | | | | | | | | |
| Q1 <19.78 | 1,877 | 218 | 1.32 (1.22–1.44) | <0.001** | 1.17 (1.02–1.34) | 0.024* | 44 | 1.26 (0.97–1.64) | 0.084 | 174 | 1.19 (1.02–1.38) | 0.024* |
| Q2 19.78–24.54 | 1,878 | 137 | 1.21 (1.05–1.38) | 0.006** | 1.28 (1.08–1.52) | 0.005** | 38 | 1.34 (0.98–1.84) | 0.069 | 99 | 1.27 (1.05–1.55) | 0.016* |
| Q3 24.54–30.91 | 1,878 | 116 | 1.23 (0.93–1.63) | 0.141 | 1.27 (0.92–1.74) | 0.142 | 21 | 0.94 (0.49–1.81) | 0.853 | 95 | 1.38 (0.97–1.97) | 0.075 |
| Q4 ≥30.91 | 1,877 | 94 | Ref. | | Ref. | | 23 | Ref. | | 71 | Ref. | |
| P for trend | | | <0.001 | | <0.001 | | | 0.136 | | | 0.001 | |

**Notes.**
[a] OR[a] univariate analyses.
[b] OR[b] adjusted for weight gain during pregnancy, father's BMI, total energy intake, monthly income per capita, maternal education level, smoking, maternal employ, multivitamin supplement, gestational hypertension, history of premature birth, reproductive history and dietary zinc intake.
** P-value < 0.01.
* P-value < 0.05.

during pregnancy was observed. The synergy index was 2.11 (S = 2.11; 95% CI [1.12–3.95]) during pregnancy; the relative excess risk ratio attributable to additive interaction was 0.97 (RERI = 0.97; 95% CI [0.34–1.60]); and 34% (AP = 0.34; 95% CI [0.14–0.54]) of the LBW risk was attributable to the interaction of low maternal dietary iron and zinc intake. At the same time, there was a multiplicative interaction between low dietary iron intake and low dietary zinc intake during pregnancy (OR = 1.82; 95% CI [1.45–2.29]). Similar findings were obtained after stratifying by the trimesters of pregnancy (Table 6).

Table 3 Maternal dietary iron intake categorized by RNIs and the odds of LBW.

| Dietary iron intake (mg/d) | NBW (7,510) | LBW (n = 565) | | | | | Term-LBW (37≥weeks) | | | Preterm-LBW (<37 weeks) | | |
|---|---|---|---|---|---|---|---|---|---|---|---|---|
| | | Cases | OR[a] (95% CI) | P-value | OR[b] (95% CI) | P-value | Cases | OR[b] (95% CI) | P-value | Cases | OR[b] (95% CI) | P-value |
| Preconception | | | | | | | | | | | | |
| Met the RNI | 3,829 | 233 | Ref. | | Ref. | | 57 | Ref. | | 176 | Ref. | |
| Below the RNI | 3,681 | 332 | 1.48 (1.25–1.76) | <0.001** | 1.26 (1.03–1.54) | 0.024* | 69 | 1.06 (0.72–1.57) | 0.770 | 263 | 1.35 (1.08–1.69) | 0.009** |
| During pregnancy | | | | | | | | | | | | |
| Met all the RNI | 2,199 | 118 | Ref. | | Ref. | | 29 | Ref. | | 89 | Ref. | |
| Below 1 trimester | 1,697 | 99 | 1.09 (0.83–1.43) | 0.535 | 1.08 (0.79–1.47) | 0.627 | 19 | 0.87 (0.45–1.66) | 0.676 | 80 | 1.22 (0.86–1.73) | 0.263 |
| Below 2 trimester | 1,428 | 114 | 1.22 (1.07–1.39) | 0.003** | 1.19 (1.01–1.41) | 0.041* | 23 | 1.12 (0.81–1.55) | 0.497 | 91 | 1.23 (1.02–1.48) | 0.029* |
| Below 3 trimester | 2,186 | 234 | 1.26 (1.17–1.36) | <0.001** | 1.13 (1.01–1.27) | 0.037* | 55 | 1.16 (0.93–1.45) | 0.190 | 179 | 1.14 (1.01–1.30) | 0.041* |
| First trimester | | | | | | | | | | | | |
| Met the RNI | 5,203 | 320 | Ref. | | Ref. | | 69 | Ref. | | 251 | Ref. | |
| Below the RNI | 2,307 | 245 | 1.73 (1.45–2.05) | <0.001** | 1.25 (1.02–1.55) | 0.037* | 57 | 1.44 (0.96–2.17) | 0.078 | 188 | 1.27 (1.00–1.60) | 0.047* |
| Second trimester | | | | | | | | | | | | |
| Met the RNI | 3,937 | 226 | Ref. | | Ref. | | 49 | Ref. | | 177 | Ref. | |
| Below the RNI | 3,573 | 339 | 1.65 (1.39–1.97) | <0.001** | 1.29 (1.04–1.60) | 0.020* | 77 | 1.40 (0.92–2.14) | 0.118 | 262 | 1.28 (1.01–1.64) | 0.046* |
| Third trimester | | | | | | | | | | | | |
| Met the RNI | 2,279 | 120 | Ref. | | Ref. | | 30 | Ref. | | 90 | Ref. | |
| Below the RNI | 5,231 | 445 | 1.62 (1.32–1.99) | <0.001** | 1.27 (0.98–1.63) | 0.066 | 96 | 1.09 (0.67–1.78) | 0.729 | 349 | 1.35 (1.02–1.80) | 0.038* |

Notes.

[a] OR[a] univariate analyses.

[b] OR[b] adjusted for weight gain during pregnancy, father's BMI, total energy intake, monthly income per capita, maternal education level, smoking, maternal employ, multivitamin supplement, gestational hypertension, history of premature birth, reproductive history and dietary zinc intake.

** $P$-value < 0.01.

* $P$-value < 0.05.

# DISCUSSION

Pregnancy is a critical period for both the short- and long-term health effects of mothers and infants. To meet the needs of their own and fetal growth and development during pregnancy, pregnant women have an increased demand for various nutrients. However, in developing countries, women tend to pay more attention to the intake of macronutrients such as protein during pregnancy than some trace elements, increasing the risk of LBW (*Haider & Bhutta, 2017*; *Parodi et al., 2018*). It has been shown (*Guilbert, 2003*; *Prasad, 2003*) that more than 2 billion people worldwide are iron-deficient, and billions more are exposed to zinc deficiency, the vast majority of which occur in developing countries. Causes of micronutrient deficiencies include inadequate intake, poor absorption, impaired utilization, excessive loss, increased physiological requirements, or a combination of these factors. Inadequate intake could be the main cause of deficiency in developing countries (*Gibson, 2006*). A nutritional survey of pregnant women in western China implied (*Mao et al., 2020*) that the deficiency ratio of dietary iron, zinc, iodine, copper, and selenium was higher during pregnancy ($P < 0.05$). Iron plays a critical role in energy metabolism, regulation of DNA synthesis, cell proliferation and differentiation, and oxygen transport. Nearly half of pregnant women in the second and third trimesters consume less than their average dietary iron requirements (estimated average requirement, EAR). Iron

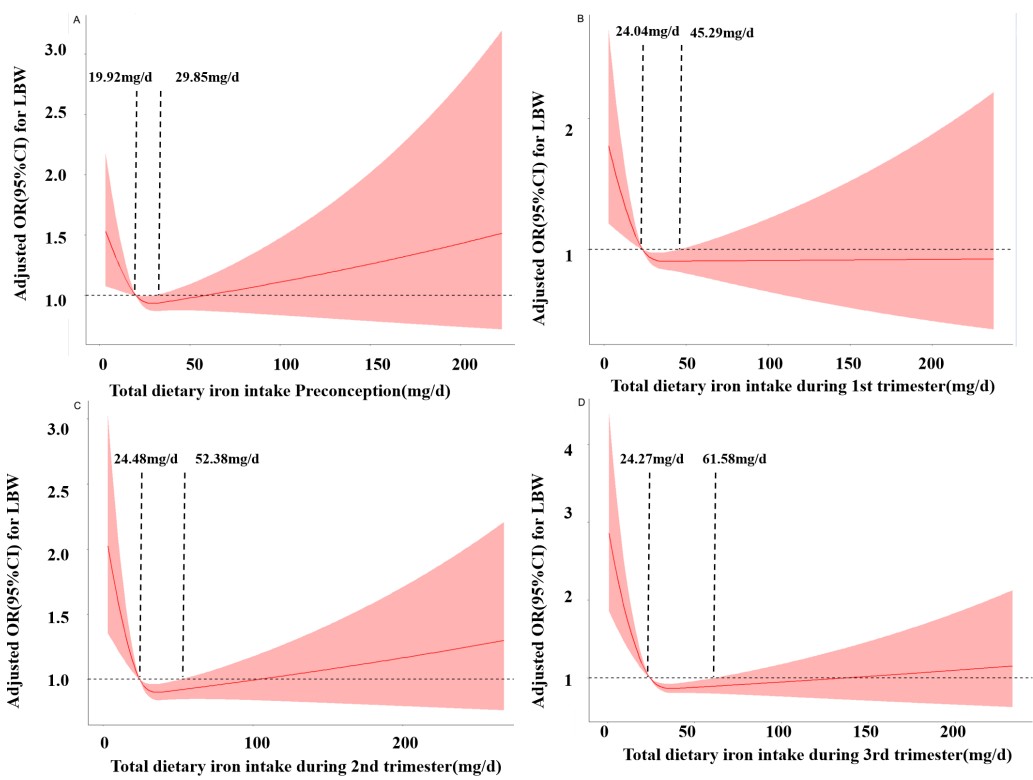

**Figure 1  Restricted cubic spline models of LBW odds associated with iron intake preconception (A), at the first trimester (B), at the second trimester (C), and at the third trimester (D).**

absorption during pregnancy is a complex process and generally increases with the increase of gestational age. The incidence of iron deficiency anemia is high in women during pregnancy, with the highest prevalence in the second and third trimesters. Therefore, the WHO recommends iron supplementation during pregnancy to prevent birth related adverse outcomes in the future (*Rahman et al., 2021*). While prenatal iron supplementation has a beneficial effect on women with anemia, it has a controversial effect on women without anemia (*Ribot et al., 2012*); namely, in women with normal hemoglobin levels during pregnancy, daily iron supplementation may be harmful to fetal growth (*Aranda et al., 2013*). A cohort study (*Shastri et al., 2015*) revealed that women with the highest dose (>39.2 mg/day) of supplemental iron intake had an increased risk of term LBW compared with those with the lowest dose (≤36.6 mg/day) (RR = 1.89; 95% CI [1.26–2.83]). Pregnant women should not take iron supplementation if their hemoglobin levels exceed 110 g/L during the first and third trimesters and 105 g/L during the second trimester. Excessive or insufficient dietary iron in mothers can both increase the risk of low birth weight. Excessive iron can lead to endocrine disorders, such as hypothyroidism, elevated HbA1c levels and decreased adrenocorticotropic hormone levels, thereby affecting bone health (*Chen et al., 2024*; *Yang et al., 2020*). In plant-based diets, iron exists in the form of inorganic non-haem iron, which is less bioavailable than haem iron from animal foods sources, and this results in lower total iron absorption from vegetarian diets (*Hurrell & Egli, 2010*). Studies have

**Table 4  Associations of maternal dietary zinc intake with the odds of LBW.**

| Dietary zinc intake (mg/d) | NBW (7,510) | LBW (n = 565) | | | | | Term-LBW (37≥weeks) | | | Preterm-LBW (<37 weeks) | | |
|---|---|---|---|---|---|---|---|---|---|---|---|---|
| | | Cases | OR[a] (95% CI) | P-value | OR[c] (95% CI) | P-value | Cases | OR[c] (95% CI) | P-value | Cases | OR[c] (95% CI) | P-value |
| Preconception | | | | | | | | | | | | |
| Q1 <5.34 | 1,878 | 232 | 1.29 (1.20–1.40) | <0.001[**] | 1.05 (0.95–1.17) | 0.360 | 47 | 0.93 (0.76–1.14) | 0.483 | 185 | 1.08 (0.96–1.22) | 0.208 |
| Q2 5.34–6.79 | 1,877 | 123 | 1.07 (0.94–1.23) | 0.306 | 0.96 (0.82–1.13) | 0.617 | 31 | 0.81 (0.60–1.08) | 0.160 | 92 | 0.98 (0.82–1.17) | 0.824 |
| Q3 6.79–8.52 | 1,878 | 103 | 0.96 (0.73–1.27) | 0.785 | 0.98 (0.72–1.33) | 0.900 | 19 | 0.64 (0.35–1.17) | 0.147 | 84 | 1.02 (0.73–1.43) | 0.115 |
| Q4 ≥ 8.52 | 1,877 | 107 | Ref. | | Ref. | | 29 | Ref. | | 78 | Ref. | |
| P for trend | | | <0.001 | | 0.019 | | | 0.975 | | | 0.005 | |
| During pregnancy | | | | | | | | | | | | |
| Q1 <6.71 | 1,877 | 243 | 1.39 (1.27–1.51) | <0.001[**] | 1.22 (1.01–1.46) | 0.034[*] | 51 | 0.92 (0.64–1.33) | 0.654 | 192 | 1.34 (1.09–1.63) | 0.004[**] |
| Q2 6.71–8.40 | 1,878 | 140 | 1.15 (1.05–1.26) | 0.002[**] | 1.10 (0.91–1.32) | 0.315 | 29 | 0.81 (0.52–1.24) | 0.342 | 111 | 1.09 (0.88–1.34) | 0.442 |
| Q3 8.40–10.39 | 1,877 | 90 | 0.88 (0.79–1.02) | 0.888 | 0.93 (0.67–1.28) | 0.661 | 22 | 0.88 (0.47–1.65) | 0.689 | 68 | 0.94 (0.65–1.36) | 0.742 |
| Q4 ≥ 10.39 | 1,878 | 92 | Ref. | | Ref. | | 24 | Ref. | | 68 | Ref. | |
| P for trend | | | <0.001 | | <0.001 | | | 0.658 | | | <0.001 | |
| First trimester | | | | | | | | | | | | |
| Q1 <6.46 | 1,877 | 231 | 1.34 (1.23–1.45) | <0.001[**] | 1.02 (0.85–1.17) | 0.808 | 47 | 0.81 (0.59–1.10) | 0.185 | 184 | 1.21 (0.99–1.48) | 0.063 |
| Q2 6.46–8.13 | 1,878 | 137 | 1.19 (1.04–1.36) | 0.012[*] | 1.11 (0.93–1.33) | 0.253 | 31 | 0.80 (0.55–1.19) | 0.257 | 106 | 1.07 (0.89–1.27) | 0.456 |
| Q3 8.13–10.09 | 1,878 | 100 | 1.03 (0.77–1.37) | 0.838 | 0.91 (0.67–1.24) | 0.547 | 23 | 0.83 (0.44–1.54) | 0.56 | 77 | 0.94 (0.66-1.34) | 0.732 |
| Q4 ≥ 10.09 | 1,877 | 97 | Ref. | | Ref. | | 25 | Ref. | | 72 | Ref. | |
| P for trend | | | <0.001 | | 0.055 | | | 0.855 | | | 0.002 | |
| Second trimester | | | | | | | | | | | | |
| Q1 <6.76 | 1,877 | 244 | 1.38 (1.27–1.49) | <0.001[**] | 1.14 (0.95–1.37) | 0.160 | 50 | 0.95 (0.67–1.34) | 0.772 | 194 | 1.24 (1.01–1.51) | 0.036[*] |
| Q2 6.76–8.50 | 1,878 | 135 | 1.20 (1.05–1.37) | 0.009[**] | 1.07 (0.90–1.28) | 0.451 | 28 | 0.82 (0.54–1.25) | 0.354 | 107 | 1.12 (0.92–1.37) | 0.264 |
| Q3 8.50–10.57 | 1,877 | 92 | 0.98 (0.73–1.32) | 0.889 | 0.93 (0.68–1.29) | 0.656 | 24 | 0.91 (0.49–1.69) | 0.764 | 68 | 0.94 (0.65–1.35) | 0.740 |
| Q4 ≥ 10.57 | 1,878 | 94 | Ref. | | Ref. | | 24 | Ref. | | 70 | Ref. | |
| P for trend | | | <0.001 | | 0.002 | | | 0.866 | | | <0.001 | |
| Third trimester | | | | | | | | | | | | |
| Q1 <6.74 | 1,877 | 243 | 1.38 (1.27–1.50) | <0.001[**] | 1.20 (1.01–1.43) | 0.040[*] | 50 | 0.99 (0.71–1.38) | 0.952 | 193 | 1.28 (1.05–1.55) | 0.013[*] |
| Q2 6.74–8.49 | 1,878 | 146 | 1.25 (1.10–1.43) | 0.001[**] | 1.15 (0.98–1.36) | 0.094 | 31 | 0.92 (0.62–1.36) | 0.678 | 115 | 1.21 (1.00–1.45) | 0.044[*] |
| Q3 8.49–10.56 | 1,878 | 83 | 0.89 (0.66–1.21) | 0.459 | 0.88 (0.64–1.23) | 0.443 | 22 | 0.94 (0.50–1.75) | 0.846 | 61 | 0.87 (0.60–1.26) | 0.462 |
| Q4 ≥ 10.56 | 1,877 | 93 | Ref. | | Ref. | | 23 | Ref. | | 70 | Ref. | |
| P for trend | | | <0.001 | | <0.001 | | | 0.511 | | | <0.001 | |

**Notes.**

[a] OR[a] univariate analyses.

[c] OR[c] adjusted for weight gain during pregnancy, father's BMI, total energy intake, monthly income per capita, maternal education level, smoking, maternal employ, multivitamin supplement, gestational hypertension, history of premature birth, reproductive history and dietary iron intake.

[**] P-value < 0.01.

[*] P-value < 0.05.

shown that the bioavailability of iron can be enhanced by taking iron supplements orally, adjusting the diet, including increasing the intake of iron-rich foods such as meat, as well as through techniques like sprouting and fermentation (*Abbaspour, Hurrell & Kelishadi, 2014*).

Zinc is an essential trace element and is very important for the normal development of the fetus during pregnancy. Zinc deficiency has become a major nutritional and health problem worldwide (*Roohani et al., 2013*). Considering the differences in ethnicity, living

**Table 5  Maternal dietary zinc intake categorized by RNIs and the odds of LBW.**

| Dietary zinc intake (mg/d) | NBW (7,510) | LBW (n = 565) | | | | | Term-LBW (37≥weeks) | | | Preterm-LBW (<37 weeks) | | |
|---|---|---|---|---|---|---|---|---|---|---|---|---|
| | | Cases | OR[a] (95% CI) | p-value | OR[c] (95% CI) | p-value | Cases | OR[c] (95% CI) | p-value | Cases | OR[c] (95% CI) | p-value |
| Preconception | | | | | | | | | | | | |
| Met the RNI | 2,818 | 158 | Ref. | | Ref. | | 37 | Ref. | | 121 | Ref. | |
| Below the RNI | 4,692 | 407 | 1.44 (1.12–1.86) | <0.001[**] | 1.11 (0.89–1.39) | 0.362 | 89 | 0.97 (0.62–1.51) | 0.382 | 318 | 1.20 (0.93–1.55) | 0.162 |
| During pregnancy | | | | | | | | | | | | |
| Met all the RNI | 2,120 | 110 | Ref. | | Ref. | | 28 | Ref. | | 82 | Ref. | |
| Below 1 trimester | 519 | 30 | 1.11 (0.74–1.69) | 0.624 | 1.00 (0.64–1.57) | 1.000 | 8 | 0.94 (0.41–2.18) | 0.884 | 22 | 1.02 (0.61–1.70) | 0.940 |
| Below 2 trimester | 388 | 21 | 1.02 (0.80–1.30) | 0.873 | 0.93 (0.71–1.22) | 0.600 | 4 | 0.75 (0.42–1.34) | 0.331 | 17 | 0.99 (0.73–1.33) | 0.948 |
| Below 3 trimester | 4,483 | 404 | 1.20 (1.12–1.29) | <0.001[**] | 1.01 (0.91–1.12) | 0.850 | 86 | 0.85 (0.69–1.04) | 0.121 | 318 | 1.08 (0.97–1.20) | 0.156 |
| First trimester | | | | | | | | | | | | |
| Met the RNI | 2,329 | 121 | Ref. | | Ref. | | 30 | Ref. | | 91 | Ref. | |
| Below the RNI | 5,181 | 444 | 1.65 (1.34–2.03) | <0.001[**] | 1.01 (0.77–1.32) | 0.942 | 96 | 0.74 (0.43–1.27) | 0.276 | 348 | 1.11 (0.82–1.51) | 0.503 |
| Second trimester | | | | | | | | | | | | |
| Met the RNI | 2,725 | 147 | Ref. | | Ref. | | 37 | Ref. | | 110 | Ref. | |
| Below the RNI | 4,785 | 418 | 1.62 (1.33–1.96) | <0.001[**] | 0.97 (0.74–1.27) | 0.824 | 89 | 0.65 (0.39–1.12) | 0.109 | 329 | 1.09 (0.81–1.47) | 0.570 |
| Third trimester | | | | | | | | | | | | |
| Met the RNI | 2,732 | 143 | Ref. | | Ref. | | 37 | Ref. | | 106 | Ref. | |
| Below the RNI | 4,778 | 422 | 1.69 (1.39–2.05) | <0.001[**] | 1.04 (0.80–1.36) | 0.773 | 89 | 0.69 (0.41–1.16) | 0.162 | 333 | 1.18 (0.88–1.59) | 0.272 |

**Notes.**

[a]OR[a] univariate analyses.

[c]OR[c] adjusted for weight gain during pregnancy, father's BMI, total energy intake, monthly income per capita, maternal education level, smoking, maternal employ, multivitamin supplement, gestational hypertension, history of premature birth, reproductive history and dietary iron intake.

[**]$P$-value < 0.01.

[*]$P$-value < 0.05.

environment, eating habits, and economic status, zinc deficiency in mainland China shows regional variability, and zinc deficiency levels in different pregnancies are not all the same (*Liang et al., 2019*). Zinc is involved in maintaining many important physiological functions in the human body, such as promoting growth and development, enhancing immunity, and promoting wound healing. Zinc is involved in many processes, such as cells (*Grzeszczak, Kwiatkowski & Kosik-Bogacka, 2020*). Zinc-binding proteins and zinc transporters are involved in the distribution and transport of zinc in the body, and they regulate and maintain zinc homeostasis in the body through various mechanisms (*Baltaci & Yuce, 2018*). Zinc deficiency can decrease the activity of Cu–Zn superoxide dismutase (SOD) (*Abolbashari et al., 2018*) and increase serum homocysteine levels (*Jing et al., 2015*), which may be the pathological basis for LBW in childbirth. The absorption and bioavailability of zinc are influenced by many factors. The two main factors affecting zinc absorption are dietary zinc intake and phytates, which together account for more than 80% of the variation in zinc absorption (*Hambidge et al., 2010*). Because of the wide range of biological functions of zinc and its importance to human health, the WHO recommends using zinc-rich foods when adequate zinc intake is not possible in the general diet. Zinc can be obtained from various sources, including food, fortified foods and supplements. Taking zinc supplements is a convenient and effective option for treating zinc deficiency and maintaining healthy levels of zinc (*Devarshi et al., 2024*). Iron deficiency and zinc deficiency

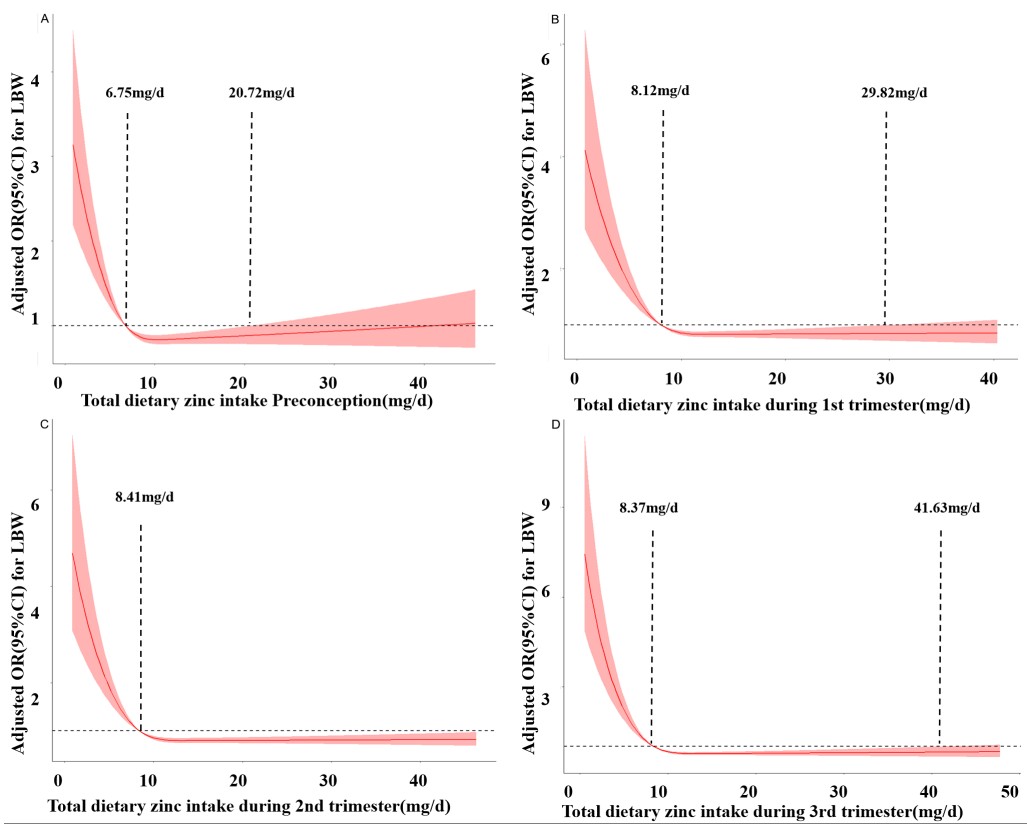

**Figure 2** Restricted cubic spline models of LBW odds associated with zinc intake preconception (A), at the first trimester (B), at the second trimester (C), and at the third trimester (D).

often occur together during pregnancy (*Nishiyama et al., 1999*). Combining iron and zinc supplementation in women with iron deficiency in early pregnancy could have more benefits than iron therapy alone. The recovery of serum ferritin in pregnant women is more effective when iron and zinc are combined 1:1 (*Saaka, Oosthuizen & Beatty, 2009*). The possible reason may be that zinc promotes the formation of IGF-1, which along with erythropoietin, is required for hematopoiesis (*Ten Have, van der Lely & Lamberts, 1997*). Additionally, it has been reported that iron loading reduces divalent metal transport-1 (DMT1) expression, resulting in reduced iron absorption. However, zinc can increase DMT1 mRNA expression (*Yamaji et al., 2001*), which may also be a potential mechanism for the interaction between dietary iron deficiency and zinc deficiency observed in this study. In this nested cohort study, we observed that women with low dietary iron intake preconception and during pregnancy had a significantly increased risk of delivering LBW infants, with a dose–response relationship, especially for preterm LBW infants. Similarly, significant inverse associations with LBW were found for lower dietary zinc intake during pregnancy. Moreover, the positive additive and multiplicative scale interactions between low dietary iron and zinc intake and LBW were observed preconception and during different trimesters of pregnancy. This is the first study that comprehensively examined

**Table 6  Interaction effects of maternal dietary iron and zinc intake on the odds of LBW.**

| Maternal dietary intake | NBW/LBW | OR[a] (95% CI) | OR[d] (95% CI) |
|---|---|---|---|
| Preconception | | | |
| High iron and high zinc | 4,142/227 | Ref. | Ref. |
| High iron and low zinc | 594/61 | 1.78 (1.29–1.59) | 1.26 (0.90–1.77) |
| Low iron and high zinc | 1,806/126 | 1.13 (1.01–1.26) | 1.12 (1.00–1.28) |
| Low iron and low zinc | 968/151 | 1.42 (1.32–1.53) | 1.24 (1.13–1.36) |

Multiplicative interaction: OR[d] (95% CI) = 1.72 (1.37–2.16), $P < 0.001$

Additive interaction: RERI (95% CI) = 0.70 (0.03–1.43), AP (95% CI) = 0.34 (0.01∼0.48),

$S$ (95% CI) = 1.61 (0.91–2.84)

| | | | |
|---|---|---|---|
| During pregnancy | | | |
| High iron and high zinc | 4,003/204 | Ref. | Ref. |
| High iron and low zinc | 1,163/100 | 1.19 (0.88–1.61) | 1.17 (0.85–1.62) |
| Low iron and high zinc | 939/57 | 1.69 (1.32–2.16) | 1.07 (0.90–1.27) |
| Low iron and low zinc | 1,405/204 | 2.85 (2.33–3.49) | 1.22 (1.09–1.37) |

Multiplicative interaction: OR[d] (95% CI) = 1.82 (1.45–2.29), $P < 0.001$

Additive interaction: RERI (95% CI) = 0.97 (0.34–1.60), AP (95% CI) = 0.34 (0.14–0.54),

$S$ (95% CI) = 2.11 (1.12–3.95)

| | | | |
|---|---|---|---|
| First trimester | | | |
| High iron and high zinc | 3,426/174 | Ref. | Ref. |
| High iron and low zinc | 977/78 | 1.57 (1.19–2.07) | 1.06 (0.75–1.49) |
| Low iron and high zinc | 1,234/77 | 1.11 (0.97–1.27) | 1.08 (0.93–1.26) |
| Low iron and low zinc | 1,873/236 | 1.35 (1.27–1.45) | 1.10 (0.99–1.23) |

Multiplicative interaction: OR[d] (95% CI) = 1.57 (1.26–1.96), $P < 0.001$

Additive interaction: RERI (95% CI) = 0.68 (0.11–1.25), AP (95% CI) = 0.27 (0.06–0.49),

$S$ (95% CI) = 1.85 (0.94–3.64)

| | | | |
|---|---|---|---|
| Second trimester | | | |
| High iron and high zinc | 4,188/221 | Ref. | Ref. |
| High iron and low zinc | 1,067/96 | 1.71 (1.13–2.19) | 1.06 (0.78–1.46) |
| Low iron and high zinc | 955/60 | 1.09 (0.94–1.26) | 1.02 (0.87–1.21) |
| Low iron and low zinc | 1,300/188 | 1.40 (1.31–1.50) | 1.20 (1.08–1.34) |

Multiplicative interaction: OR[d] (95% CI) = 1.74 (1.39–2.19), $P < 0.001$

Additive interaction: RERI (95% CI) = 0.85 (0.21–1.48), AP (95% CI) = 0.31 (0.10–0.52),

$S$ (95% CI) = 1.94 (1.04–3.62)

| | | | |
|---|---|---|---|
| Third trimester | | | |
| High iron and high zinc | 4,191/216 | Ref. | Ref. |
| High iron and low zinc | 1,066/95 | 1.73 (1.27–2.22) | 1.15 (0.84–1.59) |
| Low iron and high zinc | 981/59 | 1.08 (0.93–1.25) | 1.04 (0.88–1.23) |
| Low iron and low zinc | 1,272/195 | 1.44 (1.34–1.54) | 1.26 (1.13–1.40) |

Multiplicative interaction: OR[d] (95% CI) = 1.92 (1.53–2.42), $P < 0.001$

Additive interaction: RERI (95% CI) = 1.08 (0.43–1.73), AP (95% CI) = 0.36 (0.17–0.56),

$S$ (95% CI) = 2.20 (1.18–4.11)

**Notes.**

[a] OR[a] univariate analyses.

[d] OR[d] adjusted for weight gain during pregnancy, father's BMI, total energy intake, monthly income per capita, maternal education level, smoking, maternal employ, multivitamin supplement, gestational hypertension, history of premature birth and reproductive history.

the effects of dietary iron and zinc intake on LBW infants from the preconception period through the entire gestational period.

Few previous studies have investigated the effects of dietary iron intake preconception and during pregnancy on LBW. Most of the studies have focused on nutrient supplementation interventions and reported controversial results; thus, the evidence on whether dietary iron intake decreases the risk of LBW is still not established (*Chang, Li & Xu, 2018*). Most studies have reported that iron supplementation could decrease the risk of LBW. One prospective study (*Chang, Li & Xu, 2018*) reported that iron supplementation during the whole period of pregnancy reduced the risk of LBW (OR = 0.60; 95% CI [0.43–0.83]). However, the risk of LBW was 0.90 (95% CI [0.64–1.28]) and 0.77 (95% CI [0.58–1.03]) when iron supplementation was only given in the first trimester and in the middle-to-late pregnancy, respectively. One case–control study (*Siyoum & Melese, 2019*) in Ethiopia demonstrated that the lack of iron supplementation during pregnancy increased the risk of LBW infants (OR = 2.89; 95% CI [1.58–5.29]), and a case–control study in Spain similarly demonstrated (*Palma et al., 2008*) that 80 mg ferrous sulfate supplementation was associated with a lower risk of LBW (OR = 0.58; 95% CI [0.34–0.98]). A matched case–control study (*Yadav et al., 2020*) in Nepal indicated that iron and folic acid consumption less than 60 days before delivery raised the risk of LBW (OR = 5.47; 95% CI [2.73–10.95]). One meta-analysis (*Imdad & Bhutta, 2012*) revealed that routine daily iron supplementation during pregnancy resulted in a significant reduction of 20% in the incidence of LBW in the intervention group compared with the control group (RR = 0.80; 95% CI [0.71–0.90]). Another meta-analysis (*Haider et al., 2013*) indicated that for every 10 mg increase in iron dose/day, up to 66 mg/day, the birth weight increased by 15.1 (6.0 to 24.2) g (*P* for linear trend = 0.005), and the relative risk of LBW decreased by 3% (RR = 0.97; 95% CI [0.95–0.98]) for every 10 mg increase in dose/day (*P* for linear trend <0.001). One randomized controlled trial (*Cogswell et al., 2003*) showed that iron supplementation from enrollment to 28 weeks of gestation led to a significantly higher mean birthweight (206 ± 565 g; *P* = 0.010) and a significantly lower incidence of LBW infants (4% compared with 17%; *P* = 0.003). However, other three studies (*Imdad & Bhutta, 2012*; *Haider et al., 2013*; *Cogswell et al., 2003*; *Peña Rosas et al., 2015*; *Bánhidy et al., 2011*) observed that iron supplementation alone during pregnancy was not associated with LBW (RR = 0.78, 95% CI [0.56–1.02]), and there was no higher rate of LBW in newborns of anemic pregnant women who received iron supplementation. Studies investigating the relationship between dietary zinc intake and the risk of LBW are also conflicting (*Negandhi et al., 2014*; *Iannotti et al., 2008*; *Xie, Chen & Pan, 2001*; *Goldenberg et al., 1995*; *Zahiri Sorouri, Sadeghi & Pourmarzi, 2016*; *Osendarp et al., 2000*; *Carducci, Keats & Bhutta, 2021*; *Warthon-Medina et al., 2015*). Four of the eight studies reported that zinc supplementation reduced the risk of LBW. One case–control study (*Negandhi et al., 2014*) in India reported that low zinc intake (<5.39 mg/dL) showed a significant association with the risk of LBW. One randomized controlled trial in Peru (*Iannotti et al., 2008*) showed that zinc supplementation through pregnancy to 1 month after delivery was associated with greater weight (by 0.58 ± 0.12 kg; *P* < 0.001), but no effect was observed for linear growth. One randomized, double-blind trial (*Xie, Chen & Pan, 2001*) in western China demonstrated that zinc supplementation in

the doses of five and 10 mg/day had no benefit, whereas the 30 mg/day dose resulted in great improvements in the infants' birth weight (283.07 g, $P = 0.016$). Another randomized, double-blind placebo-controlled trial (*Goldenberg et al., 1995*) in USA demonstrated that daily zinc supplementation with 25 mg/day zinc sulfate in women with relatively low plasma zinc concentrations in early pregnancy was associated with greater infant birth weights (126 g, $P = 0.03$). However, other studies found no effect of zinc supplementation on infants' birth weight. Namely, one study reported (*Zahiri Sorouri, Sadeghi & Pourmarzi, 2016*) that 15 mg zinc supplementation daily from 16 weeks of pregnancy until delivery did not improve birth weight (3,262 g *vs.* 3,272 g, $P = 0.780$). In addition, one zinc supplementation trial indicated that no differences were noted in infants birth weight (3,267 ± 461 g *vs.* 3,300 ±498 g) by prenatal supplement type (iron + folate + zinc *vs.* iron + folate; $P > 0.05$), and there were no differences in the rates of LBW. A placebo-controlled trial in Bangladesh (*Osendarp et al., 2000*) demonstrated that supplementation with 30 mg zinc during the last two trimesters of pregnancy did not improve infant birth weights. One meta-analysis (*Carducci, Keats & Bhutta, 2021*) revealed that low or high doses of zinc supplementation during pregnancy were not significantly associated with high birth weight. Similarly, another meta-analysis (*Warthon-Medina et al., 2015*) showed that zinc supplementation during pregnancy (mean dose of 26.8 mg/day) might not attenuate the risk of LBW (RR = 0.76; 95% CI [0.52–1.11]). To the best of our knowledge, few previous studies have addressed the correlation between LBW and iron/zinc dietary intake, and there have been no previous studies of iron and zinc dietary intake interactions on LBW. Our results indicated that the simultaneous low intakes of iron and zinc preconception/during pregnancy might have an additive effect on the risk of LBW. Given the rarity of research on the combined effects of iron and zinc intakes on infant birth weight, further studies are warranted to confirm and interpret these findings. There are several strengths and limitations to be acknowledged in our study. First, information on dietary iron and zinc intake was obtained from participants' self-reports, which may have led to recall bias. Second, quantitative analysis of the interaction between dietary multivitamins and trace elements was not feasible. Thus, there could be potentially confounding effects. Therefore, strict quality control was maintained throughout the process, from questionnaire design to data entry, to minimize recall bias. Detailed information on demographic data, medical history, and lifestyle factors allowed us to adjust and control for the confounding factors. LBW and its subtypes were diagnosed based on the medical records rather than based on self-reports, which minimized the risk of potential disease misclassification; in addition, this is the first large-scale sample study to comprehensively, systematically, and deeply explore the co-effects of dietary zinc and iron intake on LBW, which has important guiding significance for the next stage of research.

## CONCLUSIONS

Our study suggests that lower intakes of iron and zinc preconception and during pregnancy may increase the risk for LBW. This study also suggests that low intakes of iron and zinc during pregnancy synergistically affect the LBW risk. These findings imply the importance

of promoting iron and zinc intakes preconception and during pregnancy to reduce the incidence of LBW in Lanzhou City. Future human studies with data on maternal biological markers, mineral intake, and genetics are warranted to confirm these findings and elucidate underlying mechanisms.

## ACKNOWLEDGEMENTS

The authors thank all the study personnel from the Gansu Provincial Maternity and Child-care Hospital (Gansu Provincial Central Hospital) for their exceptional efforts in study subject recruitment.

### Funding

This work was supported by Gansu Provincial Science and Technology Department Grant (No. 22JR5RA633), the Gansu Provincial Health Commission Research Project (No. GSWSKY2024-14; GSWSKY2024-49), and the Traditional Chinese Medicine Management Project of Gansu Province (No. GZKG-2024-43). There was no additional external funding received for this study. The funders had no role in study design, data collection and analysis, decision to publish, or preparation of the manuscript.

### Grant Disclosures

The following grant information was disclosed by the authors:
Gansu Provincial Science and Technology Department: No. 22JR5RA633.
the Gansu Provincial Health Commission Research Project: No. GSWSKY2024-14, GSWSKY2024-49.
The Traditional Chinese Medicine Management Project of Gansu Province: No. GZKG-2024-43.

### Competing Interests

The authors declare there are no competing interests.

### Author Contributions

- Liping Yang conceived and designed the experiments, performed the experiments, authored or reviewed drafts of the article, and approved the final draft.
- Zifu Wang analyzed the data, authored or reviewed drafts of the article, and approved the final draft.
- Lei Cao analyzed the data, prepared figures and/or tables, and approved the final draft.
- Yuqing Li analyzed the data, authored or reviewed drafts of the article, and approved the final draft.
- Jingyan Wang analyzed the data, prepared figures and/or tables, and approved the final draft.
- Shuyu Ding analyzed the data, prepared figures and/or tables, and approved the final draft.

- Baohong Mao conceived and designed the experiments, performed the experiments, authored or reviewed drafts of the article, and approved the final draft.

## Human Ethics

The following information was supplied relating to ethical approvals (*i.e.*, approving body and any reference numbers):

Approved was provided by the Medical Ethics Committee of Gansu Provincial Maternity and Child-care Hospital ([2018](29)).

## Data Availability

Raw data is available in the Supplemental Files.

## Supplemental Information

Supplemental information for this article can be found online at http://dx.doi.org/10.7717/peerj.19896#supplemental-information.

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
