# Peer review of "Impact of maternal iron and zinc intake on low birth weight risk: a nested case—control study"

_PeerJ, doi:10.7717/peerj.19896_

## Round 0.1 · original submission · Major Revisions

The reviewers have requested clarity on the methods used and the results thereafter.

**Language Note:** The review process has identified that the English language must be improved. PeerJ can provide language editing services - please contact us at [email protected] for pricing (be sure to provide your manuscript number and title). Alternatively, you should make your own arrangements to improve the language quality and provide details in your response letter. – PeerJ Staff

Reviewer 1 ·

Basic reporting

Please find minor editorial comments in the document attached.

Experimental design

The authors have sufficiently described their experimental design, primary aims, and questions. The investigation is relatively thorough, although some questions remain and have been asked within the PDF document attached.

Further clarification of the writing in the methods section is needed for this paper to describe when the dietary assessments were conducted, as outlined, as it is confusing to the reader unless they study the tables further down.

Recommended changes to the language about when dietary assessments were conducted are important to enable to reader to understand the effect estimates.

Validity of the findings

The findings are valid, although the spline models need to be explained. There is no information pertaining to those findings in the manuscript.

Annotated reviews are not available for download in order to protect the identity of reviewers who chose to remain anonymous.

Reviewer 2 ·

Basic reporting

1. The manuscript addresses a relevant public health question regarding the relationship between maternal micronutrient intake and the risk of low birth weight.

2. The introduction provides sufficient background, and the references are appropriate and up to date.

3. However, the quality of the English language throughout the manuscript should be improved. There are instances of inaccurate or unclear phrasing. For example, the food group “bacteria and algae” should likely be corrected to “fungi and algae.” Expressions such as “different pregnancies” need clarification, as the intended meaning is ambiguous.

4. Numerous internal referencing errors (e.g., “Error! Reference source not found.”) appear throughout the results and tables. These should be corrected before publication.

5. Figures and tables are informative, but some captions lack clarity, and the table referencing within the text must be revised for consistency.

Experimental design

1. The major concern is the inconsistency and confusion regarding the study design. The abstract refers to a “nested case-control study,” while the methods describe the implementation as a prospective cohort. Furthermore, the statistical approach, including the use of unconditional logistic regression and tests for independent and related samples, suggests a cross-sectional or unmatched case-control analysis.

2. This inconsistency undermines the rigor of the study. The authors must clearly define whether the study is a nested case-control or a cohort, and describe the parent cohort, selection criteria for cases and controls, and whether matching or pairing was performed.

3. The dietary assessment methods are generally well described, but it is unclear how and when prepregnancy dietary intake was evaluated. The current description focuses only on assessments during the three trimesters.

4. Without a clear study design and timeline, replication and interpretation are limited.

Validity of the findings

1. Although the study uses advanced statistical tools such as restricted cubic splines and interaction models, the lack of clarity in the study design raises concerns about the validity of the findings.

2. Odds ratios are presented for various time points and subgroups, but it is not evident if these analyses are appropriate for the declared design. If the study is a matched case-control, conditional logistic regression should be used. If it is a cohort, relative risks or hazard ratios may be more suitable.

3. The results, especially the interaction findings between iron and zinc intake, are interesting. However, due to methodological ambiguity, their robustness is questionable.

4. The authors must reconcile the statistical methods with a clearly defined study design to ensure internal validity.

Additional comments

The manuscript presents an important topic related to maternal and child health. The potential synergistic effects of iron and zinc intake on LBW are of interest to the scientific community. However, the lack of a clearly defined epidemiological design results in methodological inconsistencies that seriously affect the credibility of the findings. The authors must:

1. Clarify whether the study is a nested case-control or a cohort and adjust the methods and analysis sections accordingly.

2. Specify how dietary intake prior to pregnancy was assessed and justify its use in a retrospective or prospective framework.

3. Reassess the choice of statistical tests, particularly in relation to whether the data came from paired or independent observations.

4. Improve the language and fix table and figure reference errors throughout the manuscript.

---

## Round 0.2 · Minor Revisions

Reviewer 1 requests clarification on the dietary patterns underlying the RNIs, as well as a discussion of macronutrient composition and diet quality in relation to low birth weight (LBW).

Reviewer 1 ·

Basic reporting

Basic reporting is adequate and well done

Experimental design

Needs further revisions:

I remain unclear on what the diet patterns look like for this population based on which the RNIs were calculated. Given this paper is about LBW, beyond just the intake of iron and zinc, it is also important to ask questions about macronutrient composition of diets to understand why LBW was a problem in the population as it relates to zinc and iron intake. I cannot see any discussion of these important covariates, which puts this analysis into question. Some discussion of diet quality would help improve the quality of evidence being presented.

Line 227: increasing the risk of LBW

Can you speak to iron overload and how it can cause LBW?

A discussion of the bioavailability of iron and zinc in diet versus supplement form needs to be included to support gestational weight gain and LBW. This also pertains to a discussion on iron and zinc overload.

Validity of the findings

You need to speak about dietary iron and zinc in the context of diet quality. Could it be that the higher iron and zinc intakes which are predicting LBW are because of poor overall diet quality i.e., excess animal source protein and low intake of green leafy vegetables or other more nutritious sources of iron
that are essential to maintain a healthy pregnancy and adequate weight gain?

Annotated reviews are not available for download in order to protect the identity of reviewers who chose to remain anonymous.

Reviewer 2 ·

Basic reporting

The manuscript is now clearly written and well-structured. Language issues, unclear terms, and referencing errors have been resolved.

Experimental design

The authors have clearly defined the study as a non-matched nested case-control study based on a birth cohort. They now provide a detailed description of how cases and controls were selected and have justified the use of unconditional logistic regression. They also clarified that dietary intake was assessed four times, including once during the preconception period, which was not well explained in the initial version. These revisions have resolved previous methodological ambiguities.

Validity of the findings

With the clarified study design and justification for the statistical approach, the findings are now presented coherently. The restricted cubic spline analyses and interaction models are more clearly discussed, and the figures have been updated for better readability. While the study remains observational, the methodological coherence now supports the validity of the conclusions drawn.

Additional comments

The authors have provided a detailed and thoughtful rebuttal. They clarified key methodological points and improved the overall quality of the manuscript. I recommend acceptance of the revised version.

---

## Round 0.3 · accepted · Accept

All reviewers' queries were adequately addressed.

Reviewer 1 ·

Basic reporting

No concerns

Experimental design

No concerns

Validity of the findings

No concerns

Additional comments

Thank you for addressing the comments made. The paper is acceptable for publication.